# Pan-Cancer Analysis Reveals AEBP1-Collagen Co-Expression and Its Potential Role in CAF-Mediated Tumor Stiffness

**DOI:** 10.3390/ijms262311474

**Published:** 2025-11-27

**Authors:** Shohei Sekiguchi, Akira Yorozu, Megumi Watanabe, Fumika Okazaki, Satoshi Ohwada, Eiichiro Yamamoto, Takeshi Niinuma, Hiroshi Kitajima, Kazuya Ishiguro, Mitsunobu Saito, Masahiro Kai, Masashi Idogawa, Kenichi Takano, Akihiro Miyazaki, Hiroshi Ohguro, Hiromu Suzuki

**Affiliations:** 1Division of Molecular Biology, Department of Biochemistry, School of Medicine, Sapporo Medical University, Sapporo 060-8556, Japan; 2Department of Oral Surgery, School of Medicine, Sapporo Medical University, Sapporo 060-8543, Japan; 3Department of Otolaryngology-Head and Neck Surgery, School of Medicine, Sapporo Medical University, Sapporo 060-8543, Japan; 4Department of Ophthalmology, School of Medicine, Sapporo Medical University, Sapporo 060-8543, Japan; 5Division of Medical Genome Sciences, Department of Genomic and Preventive Medicine, School of Medicine, Sapporo Medical University, Sapporo 060-8556, Japan

**Keywords:** CAF, AEBP1, ACLP, collagen, 3D spheroid, tumor stiffness

## Abstract

Cancer-associated fibroblasts (CAFs) are critical components of the tumor microenvironment that promote cancer progression and immune evasion. Adipocyte enhancer-binding protein 1 gene (*AEBP1*), which encodes aortic carboxypeptidase-like protein (ACLP), has been implicated in tissue remodeling and fibrosis, yet its role in CAF biology across cancers remains poorly understood. Here, we performed a pan-cancer transcriptomic analysis using The Cancer Genome Atlas (TCGA) and found that *AEBP1* expression strongly correlates with expression of collagen family genes in the majority of solid tumors. Integration of single-cell RNA-sequencing datasets from breast and pancreatic cancers revealed that *AEBP1* is predominantly expressed in CAFs, where it is co-expressed with collagens and CAF marker genes. Functional experiments using three-dimensional (3D) spheroids composed of oral squamous cell carcinoma (OSCC)-derived CAFs showed that *AEBP1* knockdown significantly reduced spheroid stiffness without altering their morphology or size, indicating that ACLP contributes to the mechanical properties of tumor tissues. Together with earlier findings linking *AEBP1*/ACLP to reduced CD8^+^ T-cell infiltration, our results suggest that stromal *AEBP1*/ACLP enhances both extracellular matrix stiffness and immune suppression and highlights *AEBP1*/ACLP as a potential therapeutic target through which to remodel the tumor microenvironment and improve anti-tumor immunity.

## 1. Introduction

Cancer-associated fibroblasts (CAFs) are the predominant stromal cell type within the tumor microenvironment and have emerged as key regulators of tumor biology. Extensive studies have demonstrated that CAFs promote cancer progression through diverse mechanisms, including stimulation of tumor cell proliferation and invasion, angiogenesis, metabolic reprogramming, extracellular matrix (ECM) remodeling, and the induction of therapeutic resistance [1,2]. Importantly, advances in single-cell sequencing and lineage tracing have highlighted the remarkable heterogeneity of CAFs. Distinct subtypes, including myofibroblastic CAFs (myCAFs), inflammatory CAFs (iCAFs), and antigen-presenting CAFs (apCAFs), exhibit specialized molecular profiles and functional properties that influence cancer progression in different ways [3]. Beyond their established roles in stromal remodeling, CAFs have recently been recognized as crucial modulators of anti-tumor immunity. In particular, CAF-induced ECM remodeling and secretory programs can physically restrict T cell infiltration and alter T cell activation states, thereby contributing to immune evasion and resistance to immunotherapy [4]. Given their multifaceted involvement in tumor progression and immune regulation, CAFs have attracted considerable attention as therapeutic targets. Current strategies focus on either eliminating tumor-promoting CAF subsets or reprogramming their phenotypes toward tumor-suppressive functions, which reflects the growing potential of CAFs to serve as targets for precision cancer therapy [5].

We previously demonstrated that adipocyte enhancer-binding protein 1 gene (*AEBP1*) is highly expressed in the stromal compartment of colorectal cancer and promotes tumor angiogenesis [6]. The protein encoded by the *AEBP1* gene is also referred to as aortic carboxypeptidase-like protein (ACLP), a secreted ECM-associated protein [7,8,9,10]. Beyond cancer, ACLP is highly expressed in fibrotic lung tissues and has been shown to promote differentiation of myofibroblasts, highlighting its role in pathological tissue remodeling and fibrosis [11,12,13]. More recently, we reported that ACLP is abundantly expressed in CAFs from oral squamous cell carcinoma (OSCC), where it contributes to CAF activation, and that its expression is inversely correlated with intratumoral CD8^+^ T-cell infiltration [14]. This suggests ACLP may foster an immunosuppressive tumor microenvironment. Consistent with those findings, another study demonstrated that ACLP activates CAFs and promotes metastasis in pancreatic cancer through a PPARγ-dependent feedback loop [15]. Together, these data suggest that *AEBP1*/ACLP plays a critical role not only in normal fibroblasts but also in CAFs across multiple pathological contexts.

However, despite extensive studies on CAF heterogeneity and ECM remodeling, the molecular determinants that link CAF activation to mechanical stiffness across different cancers remain poorly defined. Among various ECM-associated factors, *AEBP1*/ACLP is unique in that it binds to collagen and biophysically reinforces ECM networks [16], suggesting a potential role in shaping the physical and immunological landscape of tumors. Therefore, in the present study, we performed a comprehensive pan-cancer transcriptomic analysis to characterize *AEBP1* expression patterns and its association with collagen-related genes. By integrating single-cell RNA-sequencing (scRNA-seq) data from multiple cancer types and performing functional assays using three-dimensional (3D) spheroids of CAFs, we sought to clarify whether *AEBP1*/ACLP serves as a stromal determinant that enhances tumor stiffness and contributes to immune exclusion within the tumor microenvironment.

## 2. Results

### 2.1. AEBP1-Associated Gene Expression Signatures Across Cancers

To elucidate the functional significance of *AEBP1* expression across different cancer types, we performed a Gene Ontology (GO) analysis of genes correlated with *AEBP1* expression using RNA-sequencing (RNA-seq) data from The Cancer Genome Atlas (TCGA). In breast cancer (TCGA-BRCA), *AEBP1*–correlated genes were significantly enriched in categories related to collagen organization and ECM remodeling, across the biological process (GO-BP), cellular component (GO-CC) and molecular function (GO-MF) domains (FDR < 0.05, Figure 1A). Subsequent GO-BP analysis of 32 TCGA tumor types revealed that, in nearly all cancer types, *AEBP1* expression strongly correlated with expression of genes involved in the collagen metabolic process (Figure 1B). Similarly, GO-CC analysis showed that genes related to collagen and the ECM were consistently associated with *AEBP1* across tumor types. In addition, “genes associated with the endoplasmic reticulum lumen,” “Golgi lumen,” “platelet alpha granule” and “protein complexes involved in cell adhesion” frequently showed high correlation with *AEBP1* (Figure 1C). GO-MF analysis further confirmed the strong association between *AEBP1* and collagen- or ECM-related genes and also revealed significant correlations with genes involved in “fibronectin binding,” “glycosaminoglycan binding” and “growth factor binding” (Figure 1D).

By contrast, acute myeloid leukemia (LAML) showed no collagen- or ECM-associated GO categories that significantly correlated with *AEBP1*, likely reflecting the lack of tumor stroma and the irrelevance of collagen and ECM in hematological malignancies (Figure 1B–D). Similarly, little or no enrichment of collagen- or ECM-associated GO categories was identified in thymoma (THYM) and uveal melanoma (UVM), which may also be attributable to the paucity of stromal components in these tumors (Figure 1B–D).

These findings indicate that *AEBP1* expression is closely linked to ECM remodeling programs that are conserved across a wide range of solid cancers, suggesting a fundamental stromal role for *AEBP1* in tumor biology.

### 2.2. AEBP1 Expression Strongly Correlates with Collagen and CAF Marker Gene Expression

We next examined the correlation between *AEBP1* expression and global gene expression using TCGA-BRCA data. A volcano plot revealed that many collagen family genes (e.g., *COL1A1*, *COL3A1*, *COL5A1*) were among those whose expression was the most strongly correlated with *AEBP1* expression (*p* < 0.05, Figure 2A). In addition, canonical CAF markers, including *ACTA2* (encoding α-smooth muscle actin, α-SMA), fibroblast activation protein (*FAP*) and platelet-derived growth factor receptor β (*PDGFRB*), were also strongly correlated with *AEBP1* (Figure 2A). Consistent results were observed in other tumor types, including head and neck squamous cell carcinoma (HNSC), colorectal adenocarcinoma (COADREAD) and pancreatic adenocarcinoma (PAAD), where *AEBP1* expression was strongly correlated with expression of both collagen family genes and CAF markers (Figure 2A).

To further generalize these findings, we assessed the correlation of *AEBP1* expression with CAF markers (*ACTA2*, *FAP*, and *PDGFRB*) and representative collagen family genes across 32 TCGA tumor types. This analysis demonstrated that, in the majority of cancers, *AEBP1* expression was highly correlated with expression of those genes (*p* < 0.05, Figure 2B).

Together, these findings demonstrate that *AEBP1* is transcriptionally co-regulated with the major structural and functional components of the CAF phenotype, suggesting that its expression may define a fibroblast activation program associated with ECM deposition and remodeling.

### 2.3. AEBP1 and Collagen Genes Are Co-Expressed in CAFs from Breast Cancer

To determine the cellular source of *AEBP1* expression, we analyzed scRNA-seq data from breast cancer tissues (GSE228499) [17]. Clustering and uniform manifold approximation and projection (UMAP) analyses identified 14 cell populations, among which two clusters were annotated as fibroblasts (Figure 3A). Both fibroblast subpopulations expressed *ACTA2* and *PDGFRB*, while fibroblast cluster 1 showed higher expression of *AEBP1*, *FAP* and collagen genes than fibroblast cluster 2 (Figure 3B). Visualization using UMAP and violin plots confirmed that *AEBP1* expression overlapped with collagen genes such as *COL1A1* and *COL3A1* within the same fibroblast subpopulation (Figure 3C,D). Collectively, these results indicate that in breast cancer tissue, *AEBP1* is highly expressed in CAFs and that *AEBP1*-positive CAFs co-express collagen genes. This suggests *AEBP1*-positive CAFs represent a major source of collagen production within the tumor microenvironment.

### 2.4. AEBP1 and Collagen Genes Are Co-Expressed in CAFs from Pancreatic Cancer

We further analyzed scRNA-seq data from pancreatic cancer (GSE212966) to evaluate whether the above pattern is conserved in another CAF-rich tumor type [18]. Clustering and UMAP analysis identified six fibroblast clusters, all of which exhibited high expression of *AEBP1* and collagen family genes (Figure 4A,B). Co-expression of *AEBP1* and collagen genes was further confirmed by UMAP and violin plots (Figure 4C,D).

To characterize the fibroblast heterogeneity in greater detail, we extracted fibroblast populations and performed reclustering, which revealed seven distinct clusters (Figure 5A). Clusters 0, 2, 3, 4, and 6 strongly expressed *ACTA2*, consistent with myCAFs (Figure 5B). Cluster 0 additionally expressed *MMP11* and *POSTN*, suggesting features of matrix CAFs (mCAFs) (Figure 5B) [19]. Cluster 5 expressed *IL6* and *CXCL12*, which is consistent with iCAFs (Figure 5B) [19]. Clusters 3 and 4 showed enrichment of *NOTCH3*, *MCAM* and *RGS5*, consistent with vascular CAFs (vCAFs) (Figure 5B) [19]. And cluster 6 expressed *HLA-DRA*, *HLA-DRB1* and *CD74*, corresponding to apCAFs (Figure 5B) [19].

Remarkably, *AEBP1* expression was detected in nearly all CAF subtypes, and co-expression of *AEBP1* with collagen genes (*COL1A1*, *COL3A1*) was observed across multiple fibroblast subclusters (Figure 5C,D). These data demonstrate that *AEBP1* is a core stromal gene broadly expressed among CAF populations, suggesting its potential role as a driver of ECM remodeling in solid tumors.

### 2.5. AEBP1 Enhances the Mechanical Stiffness of CAF-Derived 3D Spheroids

Given that ACLP, the protein encoded by *AEBP1*, directly interacts with collagen and enhances the stiffness of collagen fibers [16], we next assessed whether *AEBP1* contributes to the mechanical properties of CAF-derived ECM. Based on these findings, we sought to investigate the functional role of *AEBP1* in CAFs. In an earlier study, we established a three-dimensional (3D) spheroid culture model using CAFs derived from human OSCC and assessed spheroid stiffness and morphology [20]. Here, we generated 3D spheroids after knocking down *AEBP1* in OSCC-derived CAFs. The efficiency of *AEBP1* knockdown at both the mRNA and protein level using two independent siRNAs (siAEBP1-1 and siAEBP1-2) was validated in a prior study [14].

We performed knockdown experiments in two independent CAFs, followed by 3D spheroid culture. *AEBP1* knockdown did not significantly affect spheroid size (Figure 6A), nor did it markedly alter spheroid morphology as assessed by scanning electron microscopy (Figure 6B). However, *AEBP1* knockdown significantly reduced the mechanical stiffness of 3D spheroids derived from both CAFs as measured by micro-compression assays (Figure 6C). Similar results were obtained using human conjunctival fibroblasts (HconF) as normal fibroblasts, where *AEBP1* knockdown again decreased spheroid stiffness (Figure 6C, Appendix A). These results provide functional validation of the computational prediction, suggesting that *AEBP1*/ACLP directly contributes to ECM mechanics through modulation of collagen-dependent stiffness.

## 3. Discussion

In this study, we performed a pan-cancer analysis which demonstrated that *AEBP1* expression strongly correlates with collagen expression across a wide range of cancers. Previous research has consistently reported that *AEBP1* is highly expressed in various solid tumors, including gastric cancer, colorectal cancer, breast cancer, oral cancer and papillary thyroid cancer, where it promotes tumor progression, epithelial–mesenchymal transition and therapeutic resistance [21,22,23,24,25,26,27,28,29]. These findings suggest that *AEBP1* functions as a common tumor-promoting factor across diverse malignancies. By integrating scRNA-seq datasets, our study further revealed that fibroblasts represent the predominant stromal cell population expressing *AEBP1* within tumor tissues. This observation highlights the importance of CAFs as the principal source of *AEBP1* within the tumor microenvironment and provides a mechanistic link between *AEBP1* expression, collagen co-expression and stromal remodeling in cancer.

In recent years, new insights have emerged regarding the critical relationship between *AEBP1* and collagen. Several independent groups have identified bi-allelic mutations in *AEBP1* as the genetic cause of a variant of Ehlers-Danlos syndrome, a systemic connective tissue disorder characterized by defective collagen assembly and impaired ECM integrity [30,31,32,33]. Building on these genetic findings, functional studies by Vishwanath and colleagues revealed that the protein encoded by *AEBP1* (i.e., ACLP) physically interacts with collagen and enhances the stiffness, toughness, and tensile strength of collagen fibers [16]. Importantly, *AEBP1* mutations that degrade ACLP functionality abrogate those biomechanical effects, which underscores ACLP’s essential contribution to collagen fibrillogenesis and connective tissue strength [16]. Based on these insights, we hypothesized that *AEBP1*/ACLP expression in CAFs contribute to the stiffness of tumor tissues. Using our previously established CAF-derived 3D spheroid model [20], we showed that knocking down *AEBP1* prior to spheroid formation significantly reduces their mechanical stiffness as compared to spheroids derived from control CAFs. This suggests that high expression of ACLP in CAFs enhances tumor stiffness and provides a mechanistic link between stromal *AEBP1* expression, collagen remodeling and the biophysical properties of the tumor microenvironment.

Our previous work demonstrated that *AEBP1*/ACLP expression in OSCC is inversely correlated with intratumoral CD8^+^ T-cell infiltration, and that *AEBP1*/ACLP knockdown in CAFs attenuates their immunosuppressive effects on T cells [14]. Consistent with this, a recent study reported that CAF-derived AEBP1 protein directly induces T-cell dysfunction by impairing their proliferation and effector functions within the tumor microenvironment [34]. Together, these lines of evidence support a model in which *AEBP1*/ACLP acts as a stromal effector that reinforces both mechanical and immune barriers to antitumor immunity, thereby contributing to immune exclusion and resistance to immunotherapy.

From a therapeutic perspective, these findings suggest that *AEBP1*/ACLP may serve as a dual-function stromal checkpoint that integrates ECM mechanics with immune regulation. Pharmacologic targeting of *AEBP1*/ACLP or its downstream effectors may thus provide a strategy to remodel the fibrotic tumor stroma and enhance immune checkpoint inhibitor efficacy. Given that stromal stiffness and immune suppression are intimately linked in many refractory cancers, interventions that reduce *AEBP1*/ACLP activity could potentially restore tissue elasticity and improve cytotoxic T-cell access to tumor nests.

This study has several limitations that should be acknowledged. First, the pan-cancer analysis was based solely on statistical interrogation of publicly available transcriptomic datasets. Although these results provide broad insights, further validation is needed in individual tumor types. In particular, histological and immunohistochemical analyses are needed to clarify the spatial distribution and protein-level expression of ACLP within tumor tissues. Second, our functional investigation of tissue stiffness relied exclusively on in vitro experiments using CAF-derived 3D spheroid cultures. While this model recapitulates certain aspects of stromal mechanics, it does not fully reflect the complexity of in vivo tumor architecture and stromal-immune interactions. Future studies using animal models and patient-derived tissues will therefore be critical to substantiate our findings. Third, a recent study identified CKAP4 as a receptor for AEBP1 [34]. Comparative analysis of AEBP1 (ACLP) expression with CKAP4 expression in human tumor samples will be necessary to elucidate the receptor-ligand axis underlying CAF-mediated tumor remodeling and immune regulation.

## 4. Materials and Methods

### 4.1. The Cancer Genome Atlas Data Analysis

RNA-seq data from TCGA were analyzed using LinkedOmics (https://www.linkedomics.org/ (accessed on 27 April, 2025)) to identify genes whose expression correlated with *AEBP1* expression [35]. Pearson’s correlation test was applied to calculate correlation coefficients between *AEBP1* expression and the expression of all other genes. GO analysis was performed using the Gene Set Enrichment Analysis (GSEA) tool implemented in LinkedOmics. The resulting data were visualized using the ggplot2 package in R v4.5.0.

### 4.2. Single-Cell RNA-Sequencing Data Analysis

scRNA-seq datasets from breast cancer tissues (GSE228499) and pancreatic cancer tissues (GSE212966) were obtained from the Gene Expression Omnibus [17,18]. Analyses were performed using Seurat (v5.3.0) in R v4.5.0. Cells with fewer than 200 or more than 5000 detected genes, or with >25% mitochondrial transcript content, were excluded. After quality control, 2000 highly variable genes were selected and used for principal component analysis (PCA). Batch effects across samples were corrected with Harmony [36]. The top 20 principal components were applied for downstream analyses. Dimensional reduction was performed with UMAP, and clustering was conducted using FindNeighbors and FindClusters with a resolution of 0.5. Cell type annotation was carried out using SingleR (v 2.10.0) with the Human Primary Cell Atlas as a reference.

### 4.3. Three Dimensional Cell Culture Experiments

CAFs were obtained from tongue OSCC patients and cultured as described [14]. Human conjunctival fibroblasts (HconF) were obtained and cultured as described [37]. Transfection was carried out using the TransIT-X2 Dynamic Delivery System (Mirus Bio, Madison, WI, USA) with 100 pmol of Silencer Select siRNAs targeting *AEBP1* (siRNA1, s1145; siRNA2, s1146) or the Silencer Select Negative Control No. 1 siRNA (Thermo Fisher Scientific, Waltham, MA, USA), as described previously [14]. Three-dimensional spheroid drop culture of the cells was then performed as described previously [20]. Morphological analysis of 3D spheroids using phase contrast microscopy and scanning electron microscopy were performed as described [38]. The mechanical stiffness of the 3D spheroids was measured using MicroSquisher (CellScale, Waterloo, ON, Canada) as described [39]. This study was approved by the Institutional Review Board at Sapporo Medical University (No. 322–38, approved on 14 May 2020). Informed consent was obtained from all subjects involved in the study.

### 4.4. Statistical Analysis

Statistical analyses of continuous variables were performed using one-way ANOVA. A two-sided *p*-value < 0.05 was considered statistically significant. GraphPad Prism version 5 (GraphPad Software, La Jolla, CA, USA) was used for data analyses.

## 5. Conclusions

In conclusion, our study revealed that *AEBP1* expression is strongly correlated with collagen expression across multiple cancers and that *AEBP1* is predominantly expressed by CAFs, where it contributes to the enhancement of tumor stiffness. This suggests that *AEBP1*/ACLP represents a promising therapeutic target through which to remodel the tumor microenvironment and enhance anti-tumor immunity. Future research will be essential to validate its role in vivo, clarify the underlying mechanisms, and explore the potential of targeting *AEBP1*/ACLP in combination with existing immunotherapies to improve cancer treatment outcomes.

## Figures and Tables

**Figure 1 ijms-26-11474-f001:**
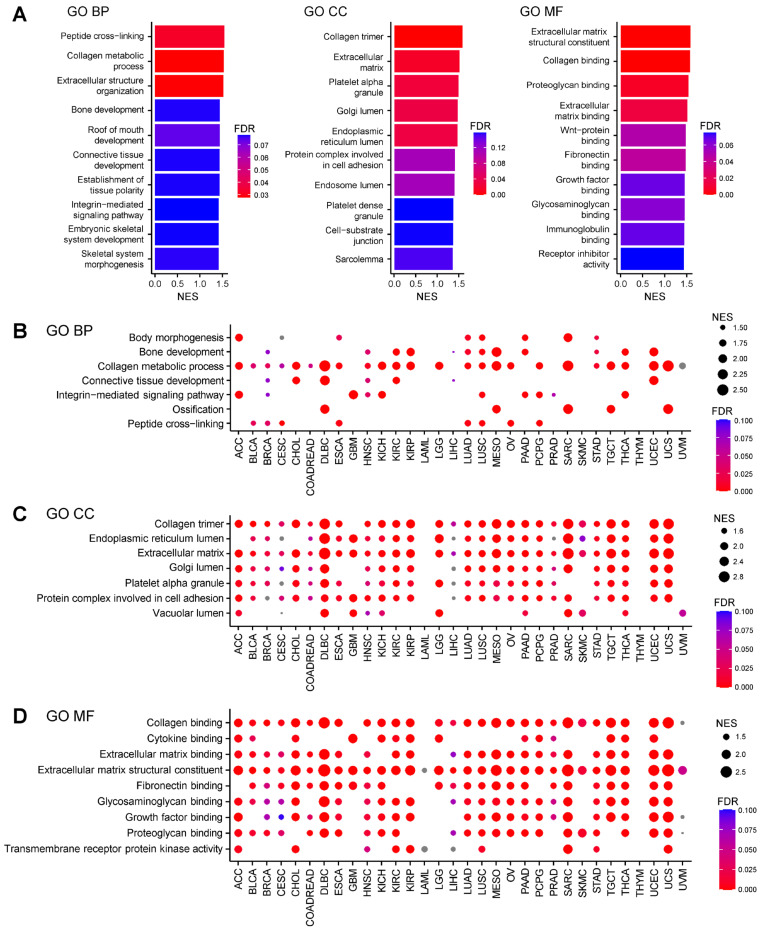
Characteristics of gene expression correlated with *AEBP1* expression. (**A**) Summary of a Gene Ontology (GO) analysis of genes whose expression correlated with *AEBP1* expression in breast cancer (TCGA-BRCA). Left, GO biological process (GO-BP); middle, GO cellular component (GO-CC); right, GO molecular function (GO-MF). NES, normalized enrichment score; FDR, false discovery rate. (**B**) GO-BP analysis of gene expression correlated with *AEBP1* expression across the indicated TCGA tumor types. Grey dots indicate GO categories in which no significant enrichment was detected (i.e., FDR outside the color scale range). ACC, adrenocortical carcinoma; BLCA, bladder urothelial carcinoma; BRCA, breast invasive carcinoma; CESC, cervical squamous cell carcinoma and endocervical adenocarcinoma; CHOL, cholangiocarcinoma; COADREAD, colorectal adenocarcinoma; DLBC, diffuse large B-cell lymphoma; ESCA, esophageal carcinoma; GBM, glioblastoma multiforme; HNSC, head and neck squamous cell carcinoma; KICH, kidney chromophobe; KIRC, kidney renal clear cell carcinoma; KIRP, kidney renal papillary cell carcinoma; LAML, acute myeloid leukemia; LGG, lower grade glioma; LIHC, liver hepatocellular carcinoma; LUAD, lung adenocarcinoma; LUSC, lung squamous cell carcinoma; MESO, mesothelioma; OV, ovarian serous cystadenocarcinoma; PAAD, pancreatic adenocarcinoma; PCPG, pheochromocytoma and paraganglioma; PRAD, prostate adenocarcinoma; SARC, sarcoma; SKCM, skin cutaneous melanoma; STAD, stomach adenocarcinoma; TGCT, testicular germ cell tumors; THCA, thyroid carcinoma; THYM, thymoma; UCEC, uterine corpus endometrial carcinoma; UCS, uterine carcinosarcoma; UVM, uveal melanoma. (**C**) GO-CC analysis of gene expression correlated with *AEBP1* expression across the indicated TCGA tumor types. (**D**) GO-MF analysis of gene expression correlated with *AEBP1* expression across the indicated TCGA tumor types.

**Figure 2 ijms-26-11474-f002:**
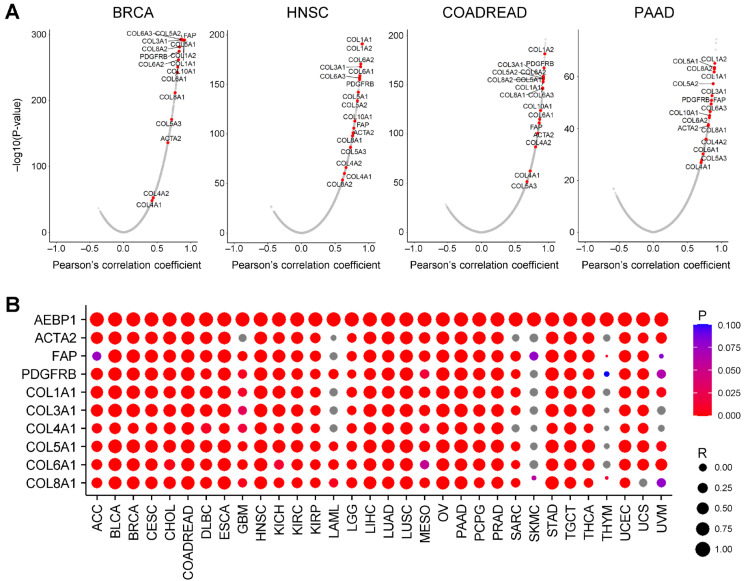
*AEBP1* expression is strongly correlated with the expression of collagen family genes. (**A**) Volcano plots showing the correlations between *AEBP1* and all genes in the indicated tumor types. (**B**) Correlation between *AEBP1* expression and expression of the indicated genes across TCGA tumor types. R, Pearson’s correlation coefficient. Grey dots indicate gene pairs in which no significant correlation was detected (i.e., *p*-value outside the color scale range).

**Figure 3 ijms-26-11474-f003:**
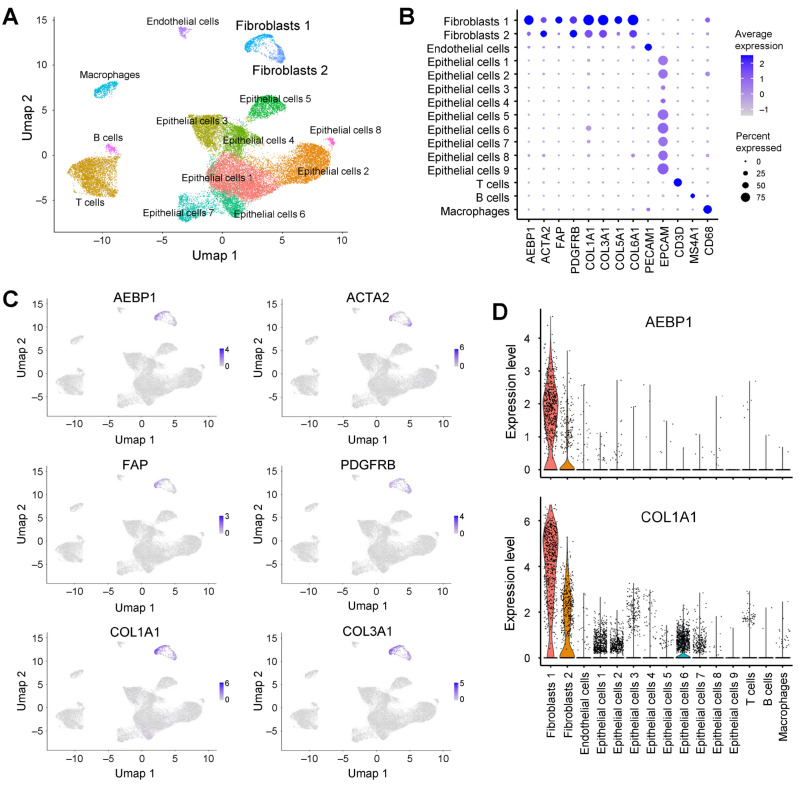
*AEBP1* is preferentially expressed in fibroblasts in breast cancer. (**A**) UMAP plot of scRNA-seq data from breast cancer tissues (GSE228499) annotated by cell type. (**B**) Dot plot showing the expression levels of *AEBP1*, CAF markers, collagen genes and representative marker genes in each cell type. (**C**) UMAP plots showing the expression levels of the indicated genes. (**D**) Violin plots showing *AEBP1* and *COL1A1* expression levels across the indicated cell types.

**Figure 4 ijms-26-11474-f004:**
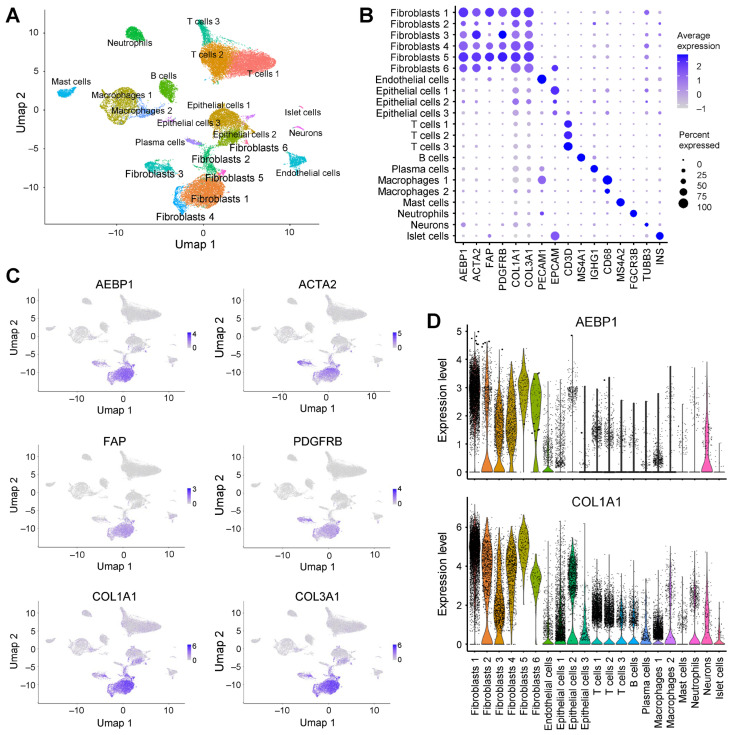
*AEBP1* is preferentially expressed in fibroblasts in pancreatic cancer. (**A**) UMAP plot of scRNA-seq data from pancreatic cancer tissues (GSE212966) annotated by cell type. (**B**) Dot plot showing the expression of *AEBP1*, CAF markers, collagen genes and representative marker genes for each cell type. (**C**) UMAP plots showing the expression levels of the indicated genes. (**D**) Violin plots showing *AEBP1* and *COL1A1* expression levels across the indicated cell types.

**Figure 5 ijms-26-11474-f005:**
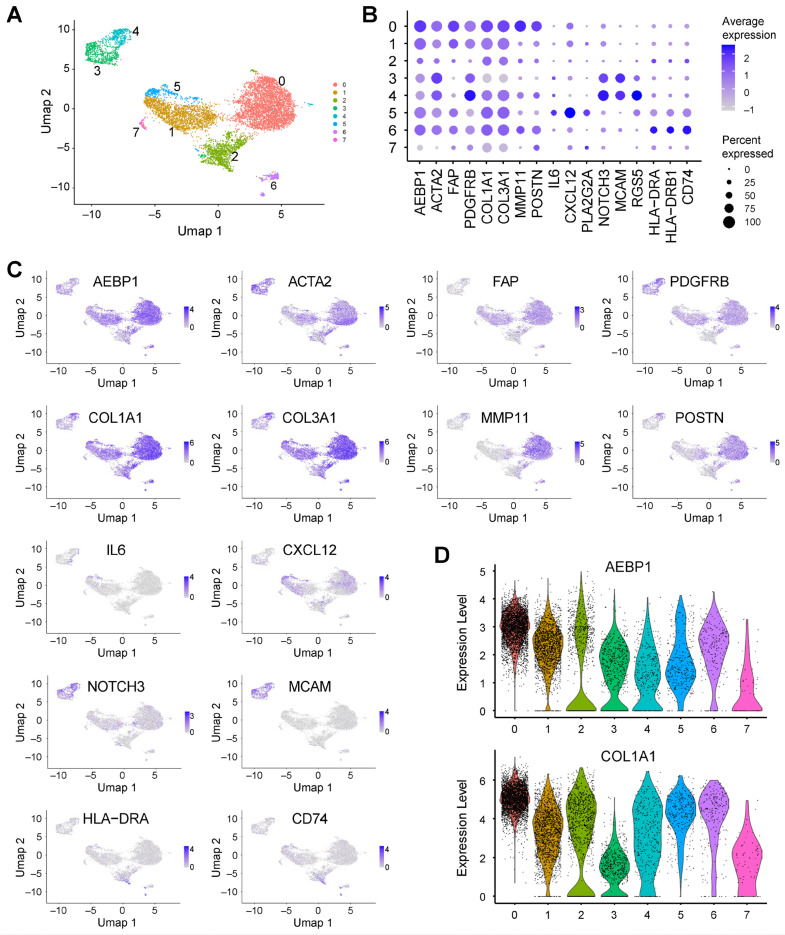
*AEBP1* expression in fibroblasts from pancreatic cancer. (**A**) UMAP plot showing fibroblasts extracted from scRNA-seq data from pancreatic cancer tissues (GSE212966). (**B**) Dot plot showing the expression of *AEBP1* and CAF marker genes across fibroblast subclusters. (**C**) UMAP plots showing the expression levels of the indicated genes. (**D**) Violin plots showing *AEBP1* and *COL1A1* expression levels across the fibroblast subclusters.

**Figure 6 ijms-26-11474-f006:**
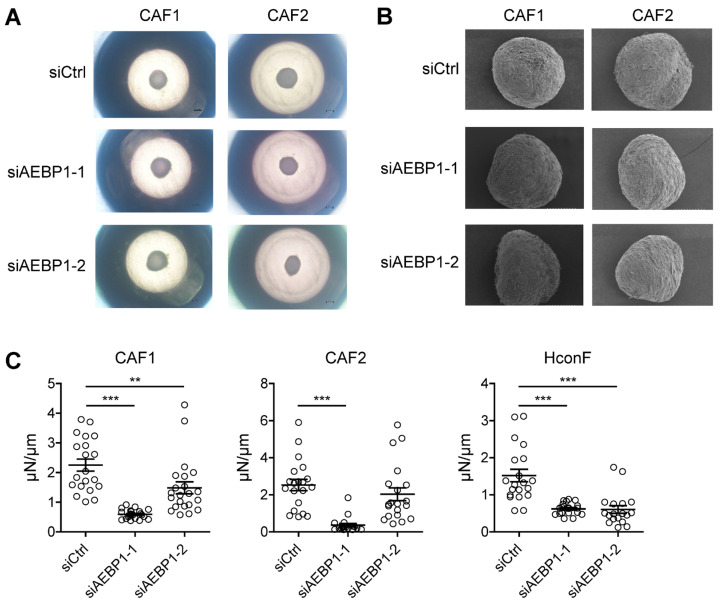
*AEBP1* knockdown attenuates the mechanical stiffness of fibroblast-derived 3D spheroids. (**A**) Representative phase-contrast micrographs of 3D spheroids derived from the indicated CAFs transfected with a control siRNA or an siRNA targeting *AEBP1* (siAEBP1-1 or siAEBP1-2). (**B**) Representative scanning electron micrographs of 3D spheroids derived from the indicated CAFs. (**C**) Mechanical stiffness of 3D spheroids obtained from the indicated CAFs and normal fibroblasts (HconF) transfected with an indicated siRNA. Each open circle represents an individual measurement, and 20 replicate measurements were collected for each condition. ** *p* < 0.01, *** *p* < 0.001.

## Data Availability

The data presented in this study are available on request from the corresponding author upon reasonable request.

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
