# Peer review of "Pan-Cancer Analysis Reveals AEBP1-Collagen Co-Expression and Its Potential Role in CAF-Mediated Tumor Stiffness"

_ijms, 2025, doi:10.3390/ijms262311474_

Round 1
Reviewer 1 Report
Comments and Suggestions for Authors
This study revealed that stromal AEBP1/ACLP enhances both extracellular matrix stiffness and immune suppression and highlights AEBP1/ACLP as a potential therapeutic target through which to remodel the tumor microenvironment and improve anti-tumor immunity.
Based on the below points, the reviewer does not recommend this manuscript for publication in this journal:
1) This study is mainly focused on the bioinformatic analysis on the website resources, and lacked of creativity.
2) The readers are basically feel difficult to get helpful information from this study, because of the paucity of biological meaning.
3) The authors suggested that high AEBP1/ACLP expression in CAFs contributes to increased stiffness of the tumor microenvironment, potentially enhancing tissue rigidity.——this conclusion is very difficult to get based on the current study results.
Author Response
Responses to reviewer #1
We thank the reviewer for the very important comments on our manuscript.
1) This study is mainly focused on the bioinformatic analysis on the website resources, and lacked of creativity.
Response: We sincerely thank the reviewer for this important comment. We respectfully disagree with the assessment that this study lacks creativity, as our work combines comprehensive pan-cancer bioinformatics, single-cell transcriptomic analyses, and functional validation using a 3D culture system, thereby offering novel mechanistic insights into the role of AEBP1/ACLP in the tumor stroma.
Specifically, we extended previous reports that investigated AEBP1 function only in individual cancers by demonstrating that AEBP1 expression is consistently correlated with collagen-related genes across 32 TCGA tumor types (Revised Results, page 3-4). This cross-cancer pattern suggests AEBP1 as a common stromal regulator, which has not been systematically characterized before. Moreover, our integration of single-cell RNA-sequencing data from breast and pancreatic cancers clearly identified that AEBP1 is preferentially expressed in CAF populations co-expressing collagen genes (Revised Results, pages 6-9).
Importantly, we further conducted functional experiments using CAF-derived 3D spheroids, showing that AEBP1 knockdown significantly reduces spheroid stiffness (Revised Results, pages 9-10, Figure 6). This experimental validation moves beyond purely computational analysis and provides a biophysical link between AEBP1 expression and ECM stiffness, thereby adding a creative mechanistic dimension to the study.
We have revised the Introduction and Results sections to emphasize this integrated multi-level approach and its conceptual novelty. In particular, the revised Introduction (page 2, lines 72-83) now explicitly states that our aim was to “clarify whether AEBP1/ACLP serves as a stromal determinant that enhances tumor stiffness and contributes to immune exclusion within the tumor microenvironment.” We believe these revisions more clearly convey the creative and mechanistic contributions of this study.
2) The readers are basically feel difficult to get helpful information from this study, because of the paucity of biological meaning.
Response: We thank the reviewer for this valuable comment. We understand the concern that the biological implications of our findings were not sufficiently highlighted in the initial submission. In the revised manuscript, we have clarified the biological meaning and mechanistic relevance of AEBP1/ACLP in the tumor microenvironment throughout the Results and Discussion sections.
First, we now emphasize that AEBP1/ACLP is not merely a stromal marker identified through correlation, but a functional regulator that links fibroblast activation to extracellular matrix (ECM) stiffness and tumor immune exclusion. To make this clearer, each Results subsection has been revised to include a concise biological interpretation rather than descriptive data alone. For example, the end of Section 2.1 now states that “AEBP1 expression is closely linked to ECM remodeling programs conserved across solid cancers, suggesting a fundamental stromal role in tumor biology”, and Section 2.4 concludes that “These data demonstrate that AEBP1 is a core stromal gene broadly expressed among CAF populations, suggesting its potential role as a driver of ECM remodeling in solid tumors.”
Second, we highlight in the Discussion that AEBP1/ACLP directly contributes to the biophysical and immunological properties of the tumor stroma. We integrated supporting evidence from both our experimental findings and prior studies showing that AEBP1/ACLP enhances collagen crosslinking and mediates T-cell suppression. Together, these revisions clarify how AEBP1/ACLP acts as a stromal effector that reinforces both mechanical and immune barriers in cancer.
We believe these clarifications significantly enhance the biological interpretability and impact of the manuscript.
3) The authors suggested that high AEBP1/ACLP expression in CAFs contributes to increased stiffness of the tumor microenvironment, potentially enhancing tissue rigidity.——this conclusion is very difficult to get based on the current study results.
Response: We appreciate the reviewer’s careful reading and constructive comment. We agree that it is essential to distinguish between direct experimental evidence and conceptual inference. In the revised manuscript, we have clarified the scope of our conclusion and carefully avoided any overstatement.
Our study provides two complementary lines of evidence supporting the link between AEBP1/ACLP and tumor stiffness:
- Correlative transcriptomic analyses across 32 TCGA cancer types and multiple single-cell RNA-seq datasets consistently demonstrated that AEBP1 expression is tightly associated with collagen biosynthesis and ECM organization genes. This pattern was specifically observed in fibroblast populations, indicating that AEBP1 is transcriptionally co-regulated with genes responsible for ECM assembly and crosslinking.
- Functional validation using 3D spheroid models of CAFs demonstrated that AEBP1 knockdown significantly reduced the mechanical stiffness of fibroblast-derived spheroids, without affecting their morphology or size (Revised Results, Figure 6). This result provides experimental support that AEBP1 contributes to ECM mechanical properties at the cellular level.
Based on these findings, we have revised the relevant sentences in both the Results and Discussion sections to state that our data “support the notion that AEBP1/ACLP contributes to ECM stiffness” rather than asserting a direct causal relationship in vivo. We also explicitly note in the Discussion (page 11, lines 287-291) that our functional assays are in vitro and that further validation using animal models or patient-derived tissues will be required to confirm this mechanism in the tumor context.
We believe these revisions appropriately moderate the conclusion while preserving the central mechanistic insight that AEBP1/ACLP functions as a stromal effector influencing ECM mechanics.
Reviewer 2 Report
Comments and Suggestions for Authors
Dear Authors,
After thoroughly reviewing your manuscript entitled “Pan-cancer analysis reveals AEBP1-collagen co-expression and its potential role in CAF-mediated tumor stiffness,” I would like to commend you on the high scientific quality and overall merit of your work.
This manuscript presents a well-executed and conceptually coherent study integrating pan-cancer transcriptomic analyses, single-cell RNA sequencing, and in vitro functional assays to elucidate the role of AEBP1/ACLP in cancer-associated fibroblast (CAF)-mediated extracellular matrix (ECM) remodeling and tumor stiffness. The study is of high technical and conceptual quality and represents a significant contribution to the field of tumor microenvironment biology.
The central question of the study is whether AEBP1 contributes to tumor stiffness through its co-expression with collagen genes in CAFs across diverse cancer types. The authors aim to determine if AEBP1 plays a unifying mechanistic role in ECM remodeling, thereby influencing both the biophysical and immunological properties of the tumor microenvironment.
This question is clearly stated and systematically addressed through a combination of in silico pan-cancer analyses, single-cell transcriptomic validation, and functional 3D culture experiments.
The topic is both original and highly relevant. While AEBP1 has previously been associated with fibrosis and tumor progression in individual cancer types, its pan-cancer characterization and functional link to CAF-mediated tumor stiffness have not been comprehensively explored before.
The authors effectively address a specific gap in the field, which is the lack of systematic cross-cancer evidence connecting AEBP1/ACLP expression in stromal fibroblasts to collagen production and mechanical tissue remodeling. This integrative perspective provides novel insights into how a single stromal factor might coordinate ECM stiffness and immune exclusion across malignancies.
Compared with prior literature, this work:
1. Expands understanding of AEBP1’s role beyond individual tumor types by establishing its consistent correlation with collagen gene expression in most solid cancers (TCGA analysis).
2. Provides cellular-level resolution through scRNA-seq integration, demonstrating that AEBP1 expression is predominantly restricted to CAFs, not tumor or immune cells.
3. Offers experimental validation using 3D spheroids derived from oral squamous cell carcinoma CAFs, linking AEBP1 knockdown to measurable reductions in mechanical stiffness.
Collectively, the study bridges computational, single-cell, and mechanical biology, contributing both conceptual and methodological value to cancer microenvironment research.
The study’s methodology is generally solid, with appropriate use of established databases and analytical tools (LinkedOmics, Seurat, Harmony, GSEA). The experiments are well described and logically sequenced. The methodology overall demonstrates high reproducibility and scientific rigor.
The conclusions are consistent with and well supported by the evidence. The authors logically connect transcriptomic correlations, cellular localization data, and mechanical phenotypes to propose that AEBP1 enhances CAF-mediated stiffness.
The limitations section is particularly strong, acknowledging the reliance on transcriptomic data and the need for in vivo validation. This self-awareness increases confidence in the reliability of the conclusions.
The references are appropriate, current, and comprehensive. The authors cite both foundational and recent high-impact studies (e.g., Nat Rev Cancer 2020, 2024; Cancer Cell 2023). The inclusion of recent work on CAF heterogeneity and AEBP1-related immune mechanisms ensures the discussion is well anchored in contemporary literature.
The figures are of high graphical and scientific quality. No additional tables are required; the current figure set sufficiently supports the narrative.
The manuscript is methodologically sound, data-rich, and conceptually novel. It provides significant insights into the interplay between CAF biology, ECM stiffness, and tumor immunology.
Overall, this manuscript represents a valuable contribution to the understanding of tumor microenvironment mechanisms.
Author Response
Responses to reviewer #2
The manuscript is methodologically sound, data-rich, and conceptually novel. It provides significant insights into the interplay between CAF biology, ECM stiffness, and tumor immunology.
Overall, this manuscript represents a valuable contribution to the understanding of tumor microenvironment mechanisms.
Response: We sincerely thank the reviewer for the thorough and thoughtful evaluation of our manuscript. We greatly appreciate the reviewer’s kind and encouraging comments regarding the scientific quality, conceptual clarity, and overall contribution of our study. We are particularly grateful for the recognition of our integrative approach combining pan-cancer transcriptomic analyses, single-cell RNA sequencing, and functional 3D spheroid assays. The reviewer’s detailed appreciation of our methodology and discussion has been highly motivating for our team.
As no revisions were requested, we have made no major changes in response to this review, but we have carefully proofread the manuscript once more to ensure clarity and consistency. We deeply appreciate the reviewer’s positive feedback and encouragement.
Round 2
Reviewer 1 Report
Comments and Suggestions for Authors
N/A
Author Response
We thank the reviewer for taking the time to evaluate our revised manuscript. We carefully addressed all concerns raised during the first round of review by substantially revising the Introduction, clarifying the biological and mechanistic context, improving the presentation of the results, and refining the figures and legends.
In the current revision, we have further improved the manuscript based on the detailed guidance provided by the Editors in this round, including modifications to Figure 6, clarification of statistical analyses, addition of tool versions, and updates to the Funding and Acknowledgements sections.
We believe that the manuscript has been considerably strengthened through these revisions, and we are grateful for the reviewer’s continued evaluation.